# WHAT MAKES FOR ROBUST MULTI-MODAL MODELS IN THE FACE OF MISSING MODALITIES?

## ABSTRACT

With the growing success of multi-modal learning, research on the robustness of multi-modal models, especially when facing situations with **missing modalities**, is receiving increased attention. Nevertheless, previous studies in this domain exhibit certain limitations, as they often lack theoretical insights or their methodologies are tied to specific network architectures or modalities. We model the scenarios of multi-modal models encountering missing modalities from an information-theoretic perspective and illustrate that the performance ceiling in such scenarios can be approached by efficiently utilizing the information inherent in non-missing modalities. In practice, there are two key aspects: (1) The encoder should be able to extract sufficiently good features from the non-missing modality; (2) The extracted features should be robust enough not to be influenced by noise during the fusion process across modalities. To this end, we introduce **U**ni-**M**odal **E**nsemble with **M**issing **M**odality **A**daptation (**UME-MMA**). UME-MMA employs uni-modal pre-trained weights for the multi-modal model to enhance feature extraction and utilizes missing modality data augmentation techniques to better adapt to situations with missing modalities. Apart from that, UME-MMA, built on a late-fusion learning framework, allows for the plug-and-play use of various encoders, making it suitable for a wide range of modalities and enabling seamless integration of large-scale pre-trained encoders to further enhance performance. And we demonstrate UME-MMA's effectiveness in audio-visual datasets (e.g., AV-MNIST, Kinetics-Sound, AVE) and vision-language datasets (e.g., MM-IMDB, UPMC Food101).

## 1 INTRODUCTION

Recent advances in deep neural network research have led to remarkable success in various multi-modal learning tasks (Wang et al., 2016; Zhao et al., 2018; Antol et al., 2015; Radford et al., 2021b; Ramesh et al., 2021b). However, most of the multi-modal research assumes the availability of complete data for all modalities, which may not always be feasible in real-world applications (Kossinets, 2006; Suo et al., 2019). And without specific design, multi-modal models might not fully realize their potential in scenarios with missing modalities (Ma et al., 2022), which emphasizes the critical need for research focused on addressing the challenges of missing modalities in multi-modal learning.

The modality incompleteness can result from modalities being partially missing in either testing examples, training examples, or both. Ma et al. (2021) utilizes one modality to reconstruct the features of another modality. When one modality is missing, the reconstructed features can fill in the features of the missing one. However, the complexity of this method dramatically increases as the number of modalities increases; Ma et al. (2022) only focuses on the specific backbone, ViLT (Kim et al., 2021), and solely addresses the issue of missing modalities in the test set, which is restrictive; Wang et al. (2023) proposes using a shared encoder to encode different modalities, resulting in shared features. However, when there's a significant gap between modalities (such as images and text), a shared encoder often fails to produce satisfactory results (Zhang et al., 2023).

We model the scenario of missing modalities in multi-modal models within the commonly used framework of information theory (Stone, 2015). Previous work, when using information theory to model multi-view or multi-modal data, often assumes that different views carry nearly identical information (Sridharan & Kakade, 2008b; Sun et al., 2020). However, this assumption doesn't always hold true in multi-modal data. In some situations, the amount of information carried by different

modalities can vary significantly (Wang et al., 2020). In other cases, the presence of all modalities is required to make accurate predictions (Goyal et al., 2017). In considering the characteristics of multi-modal tasks, we derive the performance upper bound for multi-modal models when faced with missing modalities. This bound provides compelling evidence that fully utilizing the information of the non-missing modality to reduce the dependence between modalities is essential. And we identify two key components to achieve this in practice:

- The model should be able to extract sufficiently good features from non-missing modality. We enhance that by integrating uni-modal pre-trained models into the multi-modal framework, as uni-modal encoders often end up under-trained when directly trained jointly in a multi-modal setting (Du et al., 2023).
- The features of the non-missing modality should remain unaffected by features of the substitute for the missing modality. We employ missing-modality augmentation to better enable the non-missing modalities to adapt to situations where other modalities are absent. [1]

Based on these foundational techniques, we've developed the **U**ni-**M**odal **E**nsemble with **M**issing **M**odality **A**daptation (**UME-MMA**) method, built upon the late-fusion framework. UME-MMA stands out for its simplicity (no complex fusion methods or losses), flexibility (plug-and-play use of various encoders and adaptable to various modalities) and effectiveness (improved performance). Furthermore, it can handle scenarios with missing modalities in training or test sets. And we demonstrate its effectiveness in audio-visual datasets (e.g., AV-MNIST, Kinetics-Sound, AVE) and vision-language datasets (e.g., MM-IMDB, UPMC Food101).

## 2 RELATED WORK

**Advancements of multi-modal learning.** The advancements in deep neural networks in recent years have significantly activated many research areas of multi-modal learning (Baltrusaitis et al., 2017; Liang et al., 2021), such as multi-modal reasoning (Yi et al., 2019; Johnson et al., 2016), cross-modal retrieval (Gu et al., 2017; Radford et al., 2021a), and cross-modal translation (Ramesh et al., 2021a). A concept close to multi-modal learning is the multi-view learning (Xu et al., 2013). The theory of multi-view learning has long been studied both theoretically (Zhang et al., 2019; Tosh et al., 2021) and empirically (Sindhwani et al., 2005; Ding et al., 2021; Amini et al., 2009; Tian et al., 2019). Earlier work (Kakade & Foster, 2007; Sridharan & Kakade, 2008b) proposes the multi-view assumption: each modality suffices to predict the label. Recently, many multi-view/modal analyses adopted this assumption (Han et al., 2021; Tsai et al., 2020; Lin et al., 2021; Federici et al., 2020; Lin et al., 2022; Sun et al., 2020). However, as pointed out by (Huang et al., 2021; 2022; Du et al., 2023), this might not always hold in the multi-modal learning settings.

**Multi-modal models with missing modalities.** Modality incompleteness may arise when modalities are partially absent in testing, training examples, or in both scenarios (Ma et al., 2021). A line of work evaluates and analyzes the robustness of current multi-modal models (Yu et al., 2020; Tian & Xu, 2021; Ma et al., 2022; Rosenberg et al., 2021; Li et al., 2020; Chang et al., 2022) and finds that they are vulnerable to single-modality missing. Based on this finding, researchers continue to improve multi-modal models' performance under modality missing by designing new network architectures and fusion methods (Kim & Ghosh, 2019; Tsai et al., 2018; Ma et al., 2022; Zhang et al., 2023) and training routines (Eitel et al., 2015; Ma et al., 2021). When dealing with known missing patterns, researchers explore additional ways: data imputation through available modalities or views (Tran et al., 2017; Lin et al., 2021; Shen et al., 2020; Shang et al., 2017), or training different models for different availability of modalities (Yuan et al., 2012). Our porposed UME-MMA stands out for its simplicity, flexibility and effectiveness.

## 3 ANALYSIS AND METHOD

In this section, we start by fully considering the characteristics of multi-modal data and employ an information-theory-based framework to model it in Sec 3.1. Then, we attempt to provide the

---

[1] We assume that the model itself is unaware when a modality is missing. Even with a missing modality, the model still receives an input, but this input carries no information. More discussion can be found in Sec 3.3.

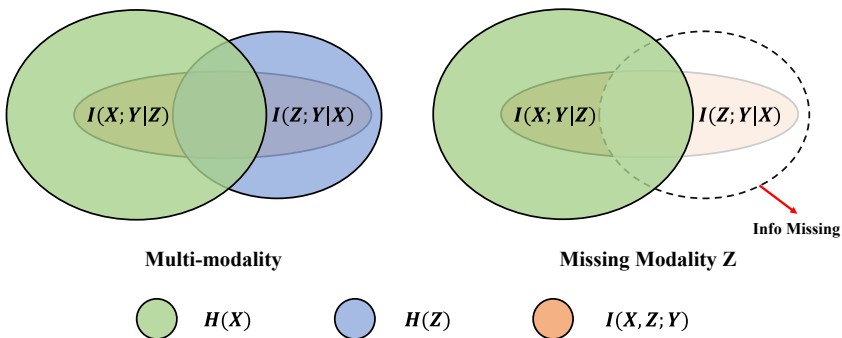

Figure 1: **Illustration of the relationship between input and target in a multi-modal task, both in normal and modality-missing scenarios**. $X$ and $Z$ are random variables representing the input of two modalities. $Y$ is the target we would like to infer. The *Info Missing* refers to the missing of modality $Z$'s information, which may prevent us from obtaining sufficient information about $Y$. Further analysis can be found in Sec 3.1 and 3.2.

performance upper bound for multi-modal models in the face of missing modalities in Sec 3.2, which underscores the importance of fully utilizing the information from the non-missing modality. In Sec 3.3, we introduce **U**ni-**M**odal **E**nsemble with **M**issing **M**odality **A**daptation (**UME-MMA**) to enhance the performance of multi-modal models in practical scenarios with missing modalities.

### 3.1 AN INFORMATION THEORY PERSPECTIVE ON MULTI-MODAL DATA MODELING

**Information theory notations.** We denote the entropy of a random variable $A$ as $H(A)$, the conditional entropy given another variable $B$ as $H(A|B)$, the mutual information between random variables $A$ and $B$ as $I(A; B)$, the conditional mutual information conditioned on random variable $C$ as $I(A; B|C)$, and the interaction information as $I(A; B; C)$.

**Multi-modal learning formulation.** We adopt the formulation for multi-modal learning problem proposed in (Huang et al., 2021). Denote the $M$-modality input space as $\mathcal{X} = \mathcal{X}_1 \times \mathcal{X}_2 \times \ldots \mathcal{X}_M$ and the target space as $\mathcal{Y}$. Each data point $(X_1, X_2, \ldots, X_M, Y)$ is sampled from an unknown distribution on $\mathcal{X} \times \mathcal{Y}$. Our goal is that, based on the random input variables $X_1, X_2, \ldots, X_M$ from $M$ modalities, we would like to infer the target $Y$. For instance, considering audio-visual action recognition (Gao et al., 2019; Feichtenhofer et al., 2016), let $X_1$ be the audio part and $X_2$ be the RGB part, and we want to infer the label $Y$, i.e., what kind of action is performed in the clip. We have also visualized the relationship between modalities and labels more intuitively in Figure 1. Note that in classification tasks, $Y$ is a discrete random variable, while in regression tasks $Y$ is continuous. And in the subsequent analysis, we will focus on the common case $M = 2$ (Feichtenhofer et al., 2016) for simplicity, and we denote the two modalities as $X \in \mathcal{X}$ and $Z \in \mathcal{Z}$ respectively. Our analysis can be extended to cases with more than two modalities at the expense of notations.

**Different modalities are not always redundant.** Previous theoretical analyses of multi-view/modal learning (Sridharan & Kakade, 2008a; Xu et al., 2013; Tosh et al., 2021) usually adopt the assumption that each view/modality is redundant in terms of predicting the target. However, this assumption does not always hold in the multi-modal learning settings. For example, the visual modality of Kinetics-400 carries significantly more label-related information than its audio modality (Wang et al., 2020). And sometimes both modalities are needed to give correct answers (Goyal et al., 2017).

**Complementary information.** If a modality requires information from another to make accurate predictions, then we consider the information from the second modality to be complementary information for the first. And in the following, we formally define complementary information within the information-theoretical framework:

**Definition 3.1** (Complementary Information). For input variables $X$, $Z$ and the target $Y$, define the complementary information provided by $X$, $Z$ as follows:

$$\Gamma_{X,Y} = I(X; Y|Z),$$
$$\Gamma_{Z,Y} = I(Z; Y|X).$$

When the target is clear from the context, we omit the $Y$ in the subscript and denote as $\Gamma_X$ and $\Gamma_Z$.

We use $I(X; Y|Z)$ to represent the conditional mutual information between variables $X$ and $Y$ given variable $Z$. It measures the amount of additional information about variable $Y$ that can be obtained by observing variable $X$, once $Z$ is known. Thus $\Gamma_X$ can characterize the complementary information of modality $Z$, which is owned by modality $X$, and similarly for $\Gamma_Z$. Hence $\Gamma_X$ together with $\Gamma_Z$ can determine the complementarity of modality $X$ and $Z$. Clearly, larger $\Gamma_X$ and $\Gamma_Z$ imply higher complementary information content.

Intuitively, if substantial complementary information exists, losing one modality can greatly impact performance of the multi-modal models. To optimize performance in such situations, it's vital to fully utilize the non-missing modality and minimize inter-modality dependence. In the next subsection, we delve into modeling this in detail.

### 3.2 THE MODEL PERFORMANCE BOUNDS UNDER MISSING-MODALITY SETTINGS

In this subsection, we first introduce Bayes error rate (Fukunaga & Hummels, 1987) to measure the model performance, which is the lowest possible error for any predictor from the multiple modalities to infer the target. Subsequently, we provide performance guarantees for the model under missing-modality testing. Note that due to space constraints, the detailed proof of the following theorem is placed in the Appendix A.

**Bayes error rate.** Formally, given two modalities $X$ and $Z$, the multi-modal Bayes errors for classification $P_{e_c}$ and regression $P_{e_r}$ are defined as follows:

$$P_{e_c} := \mathbb{E}_{x,z \sim P_{X,Z}}[1 - \max_{y \in Y} P(Y = y|x, z)],$$

$$P_{e_r} := \mathbb{E}_{x,z,y \sim P_{X,Z,Y}}[(y - \mathbb{E}[Y|x,z])^2].$$

Let us consider the missing-modality scenario and assume the modality $Z$ is missing w.l.o.g.. Then the Bayes error rates for missing-modality settings, denoted as $P_{e_c}^{\text{Miss}}$ and $P_{e_r}^{\text{Miss}}$, become:

$$P_{e_c}^{\text{Miss}} = \mathbb{E}_{x \sim P_X}[1 - \max_{y \in Y} P(Y = y|x)],$$

$$P_{e_r}^{\text{Miss}} = \mathbb{E}_{x,y \sim P_{X,Y}}[(y - \mathbb{E}[Y|x])^2].$$

Now we establish the following theoretical guarantees to quantify differences between the Bayes error rates of multi-modal and missing-modality scenarios.

**Theorem 3.2.** *For random variables $X, Z$ and discrete random variable $Y$, we have*

$$\frac{H(Y|X,Z) - \log 2}{\log |Y|} \leq P_{e_c} \leq 1 - \exp(-H(Y|X,Z)), \tag{1}$$

$$\frac{H(Y|X,Z) + \Gamma_Z - \log 2}{\log |Y|} \leq P_{e_c}^{Miss} \leq 1 - \exp(-H(Y|X,Z) - \Gamma_Z). \tag{2}$$

*For continuous random variable $Y$, if we further assume that $Y$ takes value in $[-1, 1]$, then we have*

$$P_{e_r}^{Miss} - P_{e_r} \leq \frac{1}{2}\Gamma_Z. \tag{3}$$

*Remark* 1. **The presence of complementary information leads to varied model performance between normal and missing-modality scenarios.** The **gap** between $P_{e_c}^{\text{Miss}}$ (representing the best model performance in a missing-modality setting) and $P_{e_c}$ (representing the best model performance in a normal setting) serves as an indicator of the impact of modality absence on the model's performance. In the case of the classification task, when $\Gamma_Z = 0$, signifying that $Z$ does not provide complementary information to $X$, the information from $Z$ can be entirely substituted by the information from $X$ for predicting $Y$. Consequently, in this scenario, $P_{e_c}^{\text{Miss}}$ shares the same lower and upper bounds with $P_{e_c}$, resulting in no adverse effect on the best model's performance due to modality absence. However, *complementary information is often non-zero in multi-modal data.* As $\Gamma_Z$ increases, the bound for $P_{e_c}^{\text{Miss}}$ escalates while the bound for $P_{e_c}$ remains constant. In the extreme case where $\Gamma_Z$ is sufficiently large, the lower bound of $P_{e_c}^{\text{Miss}}$ exceeds the upper bound of $P_{e_c}$, conclusively indicating that the model's performance in the missing-modality scenario is demonstrably worse than its performance in the normal scenario. And in cases where the random variable $Y$ is continuous, the analysis is similar.

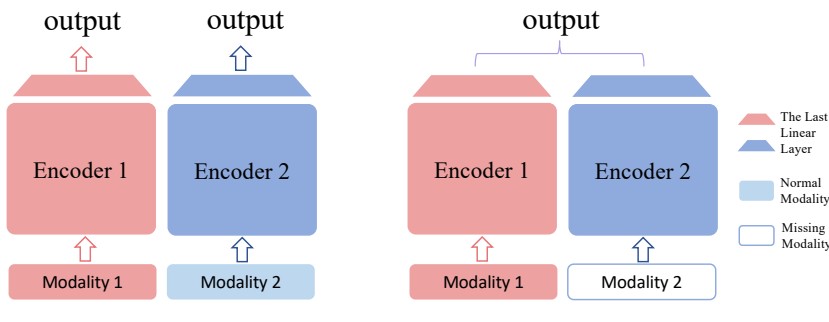

Figure 2: **The training pipeline of UME-MMA**. In the first step, we train uni-modal models using corresponding uni-modal data; Next, we obtain multi-modal output results by averaging the outputs of these pre-trained uni-modal models and use missing-modality augmentation to fine-tune the multi-modal model. This results in a multi-modal model that is robust to modality absence.

*Remark* 2. **What makes for robust multi-modal models in the face of missing modalities?** From the information theory perspective, the amount of complementary information determines how much the model's performance declines in the presence of missing modalities. The more complementary information exists, the greater the negative impact of modality missing on the model. **To reduce complementary information and ensure the model approaches its performance ceiling in scenarios of modality missing, the essence is to maximize the utilization of information from the non-missing modality.** In practice, each modality's encoder should capture all its label-related information. Specifically, for the information shared between two modalities, each encoder should be able to encode it. Otherwise, when one modality is missing, this portion of information is also lost, even though it objectively exists within the non-missing modality. However, we should also understand that in certain objective situations, both modalities need to be present to make accurate predictions (Goyal et al., 2017). In these cases, the performance decline caused by modality missing might not be mitigated by designing new training algorithms.

### 3.3 UNI-MODAL ENSEMBLE WITH MISSING MODALITY ADAPTATION (UME-MMA)

The analysis presented in the previous subsection provides us with a direction to enhance the performance of multi-modal models in scenarios with missing modalities: *fully utilizing the information of the non-missing modality to reduce the dependence between modalities*.

As illustrated in Figure 2, we consider a late-fusion multi-modal learning framework in practice: different modalities are first fed into their respective uni-modal models, producing individual outputs, and these outputs are then averaged to obtain the final results. Under this setting, we address the reduction of dependency between modalities from two perspectives:

- We initially train the uni-modal models with their respective uni-modal data and then integrate these pre-trained models into our multi-modal framework. This approach ensures the extraction of high-quality uni-modal features. Training directly in a multi-modal setting can easily lead to under-trained uni-modal encoders (Du et al., 2023).
- Next, we employ missing-modality augmentation to acclimate the multi-modal model to scenarios with modality missing. This augmentation involves randomly dropping one modality with a certain probability and replacing it with a substitute following (Hussen Abdelaziz et al., 2020). Multi-modal joint training with this technique ensures that the substitute minimally impacts the prediction of the non-missing modality. It's important to note that we never drop both modalities simultaneously. Additionally, we allocate a probability for feeding complete data into the model during training to ensure optimal performance when all modalities are available.

We name our method as **U**ni-**M**odal **E**nsemble with **M**issing **M**odality **A**daptation (**UME-MMA**), which stands out for its: (1) **Simplicity.** UME-MMA does not require any complex fusion methods, nor does it necessitate the design of complex losses; (2) **Flexibility.** It is based on a late-fusion

framework, making it adaptable to various large-scale pre-trained encoders and a wide range of input modalities. Additionally, UME-MMA is applicable even with missing modalities in the training set. In the first phase, train uni-modal models on available data. In the second, treat data points missing a modality as if they generated by missing-modality augmentation. (3) **Effectiveness.** We demonstrate its effectiveness in audio-visual datasets (e.g., AV-MNIST, Kinetics-Sound, AVE) and vision-language datasets (e.g., MM-IMDB, UPMC Food101).

**Discussion.** If we know which modality is missing, we can directly input the non-missing modality into its corresponding uni-modal model. However, in practice, a sensor may produce constant noise without meaningful information even when it's malfunctioning. In such cases, the model might not be aware that a certain modality is missing and could receive inputs with no useful information. UME-MMA is suitable for this scenario and has demonstrated strong performance.

## 4 EXPERIMENT

In this section, we first introduce the datasets used and the experimental settings. We then present the primary results of UME-MMA, followed by an ablation study of UME-MMA to better understand its working mechanism.

### 4.1 DATASETS AND EXPERIMENTAL SETTINGS

**Audio-Visual Datasets:** *1. AVE (Audio-Visual Event localization) Dataset*: Presented in Tian et al. (2018), the AVE dataset comprises 4,143 unrestricted videos, each 10 seconds in length. These videos span 28 diverse event types, including Rodents, Accordion, Mandolin, among others. Originating from the broader AudioSet collection (Gemmeke et al., 2017), the AVE dataset is split into training, validation, and testing sets with 3,339, 402, and 402 videos respectively. *2. Kinetics-Sounds Dataset*: Highlighted in Arandjelovic & Zisserman (2017), this dataset is a handpicked subset of Kinetics400, showcasing YouTube clips labeled with human actions. This collection covers 32 unique classes, encapsulating actions like playing the harmonica, tapping a pen, and shoveling snow. The dataset is partitioned into 22,728 training videos and 1,593 validation videos. *3. AV-MNIST Dataset*: The AV-MNIST dataset (Vielzeuf et al., 2018) comprises two modalities: Perturbed images and audio spectrograms. The images are derived from $28 \times 28$ PCA-projected MNIST images after removing 75% of the energy. Meanwhile, the $112 \times 112$ audio spectrograms represent spoken digits, augmented with noise. The dataset's structure mirrors that of the original MNIST, with a division of 60,000 training samples and 10,000 for testing.

**Vision-Language Datasets:** *1. MM-IMDB Dataset*: Developed by Arevalo et al. (2017), the MM-IMDB dataset merges movie plot summaries with corresponding posters for the purpose of genre classification. Given that a movie can be classified into multiple genres, this becomes a multi-label prediction task. It was crafted to address the scarcity of quality datasets for multi-modal classification. The dataset is partitioned into 15,552 training, 2,608 validation, and 7,799 testing videos; *2. UPMC FOOD101 Dataset*: Introduced by Wang et al. (2015), this dataset provides textual descriptions of 101 distinct food recipes. Extracted from meticulously selected web sources, these descriptions are complemented with images sourced from Google Image Search, which may occasionally be imprecise with regards to category accuracy. The task involves assigning the correct food label to each image-text combination. The dataset consists of 67,972 training and 22,716 testing videos.

**Experimental Settings:** As discussed in Sec 3.3, our missing-modality setup assumes that the model is unaware of the modality's absence; otherwise, the optimal approach would be to directly feed the non-missing modality into its corresponding uni-modal model. We simulate the scenario mentioned in Sec 3.3 by setting some or all of a modality's data to zero (as for the text, we simply set it to a blank space) and inputing it along with the non-missing modality into the multi-modal model. Due to space constraints, we have placed other settings in Appendix B.

### 4.2 THE EFFECTIVENESS OF UME-MMA

In this section, we demonstrate the effectiveness of UME-MMA on audio-visual datasets and vision-language datasets in scenarios where modalities are missing in the test set. Given that UME-MMA is built on a late-fusion framework, it can easily leverage more powerful large-scale pre-trained encoders

Table 1: **Top-1 test accuracy of UME-MMA and other methods on audio-visual datasets** under missing-modality and the normal settings. We also include the results of uni-modal models in the table as a reference. *Miss-V* means that visual modality is missing; *Miss-A* means audio modality is missing; and *Complete* means there are normal settings, with no modality missing; *Avg Acc* represents the average accuracy across the three settings.

| Datasets | Methods | Miss-V | Miss-A | Complete | Avg Acc |
|---|---|---|---|---|---|
| AVE | Uni-Modal Models | 35.16 | 73.30 | / | / |
| | Naive Baseline | 13.10 | 63.10 | 77.45 | 51.22 |
| | Multi-task Training | 13.60 | 58.29 | 76.94 | 49.61 |
| | Missing Detection | 23.80 | 66.25 | 70.56 | 53.54 |
| | **UME-MMA (ours)** | **36.32** | **72.12** | **79.80** | **62.75** |
| Kinetics-Sound | Uni-Modal Models | 26.76 | 62.78 | / | / |
| | Naive Baseline | 17.08 | 57.52 | 69.25 | 47.95 |
| | Multi-task Training | 21.98 | 58.02 | 70.12 | 50.04 |
| | Missing Detection | 20.90 | 62.35 | 66.71 | 49.97 |
| | **UME-MMA (ours)** | **27.34** | **62.88** | **70.23** | **53.49** |
| AV-MNIST | Uni-Modal Models | 95.84 | 97.29 | / | / |
| | Naive Baseline | 66.75 | 17.10 | 99.10 | 60.98 |
| | Multi-task Training | 92.17 | 26.32 | 99.51 | 72.67 |
| | Missing Detection | 94.45 | 95.99 | 99.74 | 96.73 |
| | **UME-MMA (ours)** | **95.24** | **97.21** | **99.77** | **97.41** |

Table 2: **Top-1 test accuracy of UME-MMA with large-scale pre-trained encoders on AVE and Kinetics-Sound** under missing-modality and normal settings. The column names of this table have the same meaning as those in Table 1.

| Datasets | Methods | Miss-V | Miss-A | Complete | Avg Acc |
|---|---|---|---|---|---|
| AVE | Uni-Modal Models | 85.60 | 88.89 | / | / |
| | Naive Baseline | 63.89 | 84.09 | **94.95** | 80.98 |
| | Multi-task Training | 54.29 | 85.35 | 93.43 | 77.69 |
| | **UME-MMA (ours)** | **86.87** | **88.13** | 94.70 | **89.90** |
| Kinetics-Sound | Uni-Modal Models | 69.62 | 84.31 | / | / |
| | Naive Baseline | 61.71 | 79.60 | 89.27 | 76.86 |
| | Multi-task Training | 62.02 | 79.10 | 88.76 | 76.63 |
| | **UME-MMA (ours)** | **71.06** | **82.93** | **89.71** | **81.23** |

to further enhance multi-modal model performance. Additionally, we also demonstrate UME-MMA is also applicable in scenarios where modalities are missing in the training set.

**UME-MMA on audio-visual datasets.** In Table 1, we demonstrate that UME-MMA performs very well on the AVE, Kinetics-Sound, and AV-MNIST datasets, both in situations with missing modalities and under normal conditions (note that the training data is complete in this experiment). We also show that UME-MMA can seamlessly integrate large-scale pre-trained encoders (Girdhar et al., 2023; Cherti et al., 2023) to further enhance its performance across various settings in Table 2. A detailed introduction to the encoders we used can be found in Appendix B due to limited space. The methods we compared in this experiment include: (1) *Naive Baseline*, which means end-to-end multi-modal late-fusion learning from scratch without carefully designed tricks, which is widely used in multi-modal learning (Du et al., 2023). (2) *Multi-task Training*. This method adds extra uni-modal linear classifiers to process the uni-modal features, which generates additional losses through the uni-modal outputs and the labels to jointly optimize the encoders. (Ma et al., 2022) use this technique to boost their model performance on missing-modality settings. (3) *Missing Detection*. Inspired by out-of-distribution (OOD) detection (Vernekar et al., 2019), we assign a new category to identify whether a modality is missing. During inference, if a uni-modal head predicts that a certain modality is missing, we rely on the prediction from the remaining modality. Otherwise, we ensemble the results from both modalities.

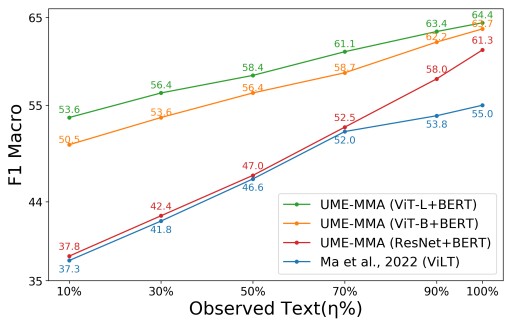 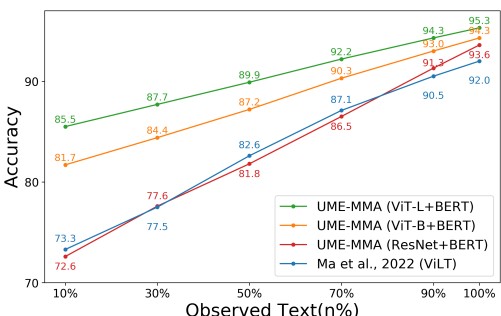

(a) Comparsion of Robustness on MM-IMDB

(b) Comparsion of Robustness on UPMC Food101

Figure 3: **The performance of UME-MMA in resisting modality missing on the MM-IMDB and UPMC Food101 datasets**. Following the settings of Ma et al. (2022), the models are trained with a combination of 100% text and 100% image data, and they are tested with a mixture of $\eta\%$ text and 100% image data. We indicate the neural network backbones used for each method. *We can see that UME-MMA also performs well in this experiment and can further enhance its performance by leveraging more powerful backbones.*

**UME-MMA on vision-language datasets.** In this experiment, we mainly follow the setting of Ma et al. (2022), using complete data during training and introducing partial text modality absence during testing. The method in Ma et al. (2022) is primarily designed for the ViLT (Kim et al., 2021) backbone (based on ViT-Base (Dosovitskiy et al., 2020) and pre-trained on vision-language datasets), aiming to enhance model performance through multi-task learning and finding the optimal fusion method. Compared to UME-MMA, this approach is more complex and would require redesigning the fusion method when aiming to leverage a stronger backbone. UME-MMA, on the other hand, allows for the seamless use of large-scale pre-trained encoders. In this experiment, we primarily followed the codebase of Kiela et al. (2019), and then implement UME-MMA using ResNet-152 (He et al., 2016), as well as ViT-Base and ViT-Large pre-trained by CLIP (Cherti et al., 2023). The results, as shown in Figure 3, indicate that when we use ResNet as the visual backbone, we have already surpassed or achieved comparable performance on both datasets compared to the method proposed in Ma et al. (2022). With the use of a stronger visual backbone, the performance has further advanced significantly. Importantly, employing a stronger encoder does not introduce any additional complexity to the UME-MMA implementation.

**UME-MMA on situations where a specific modality of data is missing in the training set.** In the above experiments, we analyze UME-MMA's performance with missing modalities in the test set. UME-MMA is also applicable to situations with missing modalities in the training set. Specifically, in the first phase of UME-MMA, each uni-modal model is trained directly on its respective and available uni-modal data. Even if one or some modalities have limited data, the result is simply that the corresponding uni-modal model trains on less data. In the second phase, during multi-modal joint training, for certain data points that might lack a specific modality, we can treat them as data generated through missing-modality augmentation, allowing the second phase to train normally. If the test set data is complete, we can even fine-tune

Table 3: **The performance of different methods on test set of MM-IMDB**. The ratios of the text modality in the training set are (10%, 20%, 100%). * indicates that the results of the method come from Ma et al. (2021).

| Method | F1 Micro↑ | | |
|---|---|---|---|
| | 10% | 20% | 100% |
| AE* | 44.8 | 50.7 | - |
| GAN* | 44.6 | 51.0 | - |
| SMIL* | 49.5 | 54.6 | - |
| UME-MMA (Res) | 45.3 | 61.0 | 66.9 |
| UME-MMA (ViT-B) | 57.5 | 65.8 | 68.4 |
| UME-MMA (ViT-L) | **60.1** | **66.3** | **69.4** |

the multi-modal model only on data with all modalities present. As shown in Table 3, UME-MMA exhibits promising results. When using ResNet as the visual encoder and having only 10% of the text modality, the performance is not as good as SMIL (Ma et al., 2021). This is primarily because SMIL's visual uni-modal performance is superior to ours (their visual uni-modal performance is 48.2,

Table 4: **Ablation study on UME-MMA with different encoders on AVE, KS and AV-MNIST**. We conduct three distinct experiments to help us understand the working principles of UME-MMA. *1. Only Missing-Aug* means incorporating missing-modality augmentation into the naive multi-modal joint training (without uni-modal pre-training). *2. Only Pre-Training* means having the multi-modal model load weights from uni-modal pre-trained models for joint training, without using missing-modality augmentation. *3. UME-MMA (Freeze Encs)* means that in the second phase of UME-MMA, we freeze the weights of the encoders and only train the final linear layer. We report the top-1 test accuracy for various methods under conditions of (video/audio) miss.

| Method | Encoder: ViT | | Encoder: CNN | | |
|---|---|---|---|---|---|
| | AVE | KS | AVE | KS | AV-MNIST |
| Only Missing-Aug | 83.1/85.6 | 65.5/80.7 | 23.8/66.3 | 20.1/61.5 | 95.2/95.1 |
| Only Pre-Training | 85.1/86.4 | 67.5/81.6 | 18.9/66.5 | 21.9/57.5 | 23.9/90.0 |
| UME-MMA (Freeze Encs) | 86.4/87.6 | 69.0/82.0 | 35.7/**72.5** | 25.3/62.5 | 94.2/**97.3** |
| UME-MMA | **86.9/88.1** | **71.1/82.9** | **36.3**/72.1 | **27.3/62.9** | **95.2**/97.2 |

Table 5: **Top-1 Test Accuracy of UME-MMA with different drop probabilities of each modality in missing-modality augmentation on AVE under modality missing and normal settings.**

| Drop Prob | 0 | 0.1 | 0.2 | 0.3 | 0.4 | 0.5 |
|---|---|---|---|---|---|---|
| Video Missing | 85.1 | 84.8 | 86.4 | 86.9 | 86.1 | 85.9 |
| Audio Missing | 86.4 | 86.6 | 87.4 | 88.1 | 88.3 | 86.1 |
| Complete | 95.0 | 94.9 | 94.4 | 94.7 | 94.9 | 93.4 |

while ours is around 44). When we use ViT-B or ViT-L as the backbone, it significantly outperforms other methods.

### 4.3 ABLATION STUDY ON UME-MMA

**Ablation on missing-modality augmentation and uni-modal pre-training.** We conduct ablation experiments for these two techniques, and the results are presented in Table 4 and Table 5:

- By solely using missing-modality augmentation, we couldn't enable the non-missing modality's encoder to extract sufficiently good features, resulting in suboptimal final outputs. This might be due to Modality Laziness that arises from multi-modal learning (Du et al., 2023).
- Merely loading the weights from uni-modal pre-trained models to the corresponding parts of the multi-modal model, without letting the multi-modal model adapt to modality missing, can also result in sub-optimal performance.
- During the second phase of training in UME-MMA, overall, fine-tuned encoders tend to perform slightly better than frozen encoders.
- From Table 5, in missing-modality augmentation, the drop probability for each modality shouldn't be too high or too low. Too high may cause training issues due to excessive information-less inputs, while too low prevents the model from adapting to missing modalities. So we settle on a 0.3 drop probability for each modality in our experiments.

## 5 CONCLUSION

This paper models the performance upper bound of multi-modal models under modality missing settings using information theory. This modeling not only establishes a theoretical foundation but also provides valuable practical insights. It emphasizes the critical role of minimizing inter-modality dependence in multi-modal models to enhance their robustness and performance in situations with missing modalities. Building upon these insights, we introduce UME-MMA, a method that leverages uni-modal pre-training to boost feature extraction capabilities and employs missing-modality augmentation to enhance adaptability. UME-MMA's key strengths lie in its simplicity, flexibility, and effectiveness. We hope this work offers new insights to the field of multi-modal learning.

## REPRODUCIBILITY STATEMENT

Firstly, in our paper, we introduce and detail the data, models, and training hyperparameters we utilized in Sec 4.1 and Appendix B. Additionally, we provide our code in the supplementary material. Given that our proposed method (in Sec 3.3) is also not complex, the reproducibility of our paper is very high.

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

# A  PROOFS

Below, we articulate the proof of Theorem 3.2 omitted in section 3.2.

*Proof.* We leverage the previous results in Feder & Merhav (1994) and Cover (1999):

$$-\log(1 - P_{e_c}) \leq H(Y|X,Z)$$
$$H(Y|X,Z) \leq \log 2 + P_{e_c} \log |Y|.$$

Combine the two inequalities and put $P_{e_c}$ in the middle:

$$\frac{H(Y|X,Z) - \log 2}{\log |Y|} \leq P_{e_c} \leq 1 - \exp(-H(Y|X,Z))$$

which is the first result in the theorem. Then we apply the results to $P_{e_c}^{Miss}$:

$$\frac{H(Y|X) - \log 2}{\log |Y|} \leq P_{e_c}^{Miss} \leq 1 - \exp(-H(Y|X)).$$

Since

$$\begin{aligned}
\Gamma_Z &= I(Z;Y|X) \\
&= I(Y;X,Z) - I(Y;X) \\
&= [H(Y) - H(Y|X,Z)] - [H(Y) - H(Y|X)] \\
&= H(Y|X) - H(Y|X,Z)
\end{aligned}$$

plug $\Gamma_Z$ into the result above, and we derive

$$\frac{H(Y|X,Z) + \Gamma_Z - \log 2}{\log |Y|} \leq P_{e_c}^{Miss} \leq 1 - \exp(-H(Y|X,Z) - \Gamma_Z)$$

which is the second result in the theorem.

Now we consider the case when $Y$ is continuous. Based on the orthogonality principle and Theorem 10 in Wu & Verdu (2012), if $Y \in [-1, 1]$, we have

$$\begin{aligned}
P_{e_r}^{\text{Miss}} - P_{e_r} &= \mathbb{E}_{x,y \sim P_{X,Y}}[(y - \mathbb{E}[Y|x])^2] - \mathbb{E}_{x,z,y \sim P_{X,Z,Y}}[(y - \mathbb{E}[Y|x,z])^2] \\
&= \mathbb{E}_{x,z,y \sim P_{X,Z,Y}}[(\mathbb{E}[Y|x] - \mathbb{E}[Y|x,z])^2] \\
&\leq \frac{1}{2} I(Z;Y|X)
\end{aligned}$$

which is the third result in the theorem. □

# B  ADDITIONAL EXPERIMENTAL SETTINGS

## B.1  EXPERIMENTAL SETTINGS ON AUDIO VISUAL DATASETS

### B.1.1  EXPERIMENTAL SETTING USING CNN AS THE BACKBONE.

**Data Preprocess.** *For AVE and Kinetics-Sound,* we adopt the data processing and augmentation used in Tian & Xu (2021). *As for AVMNIST,* we do not use data augmentation for AAV-MNIST. Note that the inputs are all scaled to the range $[-1, 1]$ (spectrogram) or $[0, 1]$ (image).

**Models and training settings.** For Kinetics-Sound and AVE, we use ResNet-18 (He et al., 2016) and AudioNet (1-D CNN) (Tian & Xu, 2021) as the backbones. We use SGD as the optimizer and set the learning rate of audio backbone and video backbone as 1e-3 and 1e-4, repsectively. We set the batch size as 48. Every ten epochs, the learning rate is reduced to 10% of its original value (total 10 epoches); For AV-MNIST, We use SGD as the optimizer and set learning rate and batch size as 1e-2 and 64 respectively. And we totally train the model for 10 epoches. During the second phase of UME-MMA, we set the learning rate to 1e-4.

### B.1.2 EXPERIMENTAL SETTING USING LARGE-SCALE PRE-TRAINED VIT AS THE BACKBONE.

**Data Preprocess.** For *images*, we adjust their size randomly to 224x224 dimensions, perform a horizontal flip, and normalize using the mean and standard deviation values from OpenCLIP. We select 3 frames at random during training and input them into the 2D network, following the approach of Peng et al. (2022); For *audio*, we follow the settings described in Girdhar et al. (2023).

**Models and training settings.** We use ViT-Base pre-trained on LAION-2B as the visual encoder (Cherti et al., 2023) and use the audio encoder of Girdhar et al. (2023) as our audio encoder. When training the models that have been pre-trained on a large scale dataset, we incorporate a linear layer to map features to labels, setting the learning rate at $1e - 5$. We use the AdamW (Loshchilov & Hutter, 2017) as the optimizer and totally train the models for 10 epoches. During the second phase of UME-MMA, we set the learning rate to 1e-5.

### B.2 EXPERIMENTAL SETTINGS ON VISION LANGUAGE DATASETS

**Data Preprocess.** For *images*, we adjust their size randomly to 224x224 dimensions, perform a horizontal flip, and normalize using the mean and standard deviation values from OpenCLIP. For *text*, We employ the *bert-base-uncased* tokenizer and the token sequence length is capped at 512.

**Models and training settings.** We use BERT-base (Devlin et al., 2018) as the text encoder and we try three different visual encoders, including ResNet-152, ViT-B and ViT-L. ViT is pre-trained on large-scale vision-language datsets (Cherti et al., 2023; Radford et al., 2021b). For training the language models and ResNet, the learning rate is 5e-5. For ViT models, it's 1e-5. During the second phase of UME-MMA, we set the learning rate to 5e-5.

