# OpenReview forum: "What Makes for Robust Multi-Modal Models in the Face of Missing Modalities?"
_ICLR.cc/2024/Conference — ICLR 2024 Conference Withdrawn Submission_

### Official Review · Reviewer_xDLC · 2023-10-26

**Soundness:** 2 fair
**Presentation:** 3 good
**Contribution:** 2 fair
**Rating:** 3
**Confidence:** 4

**Summary:**

In this paper, the authors focus on the missing modality problem in multi-modal learning field. They analyze this issue from the perspective of information theory. Based on the theoretical analysis, they conclude that the performance gap between normal case and missing modality case is caused by complementary information. Then, the UME-MMA method is proposed to overcome the missing modality case. Experiments are conducted across different modalities and models.

**Strengths:**

+ The missing modality problem is actual and worth-explored topic in multi-modal learning field.
+ The proposed method exhibits effectiveness across different settings.
+ The writing is easy to follow.

**Weaknesses:**

- Novelty of UME-MMA method is limited. Either using pre-trained uni-modal encoder or randomly dropping data of one modality during training is common tricks [A,B].
- Method design is not well-motivated. The theory conclusion is not closely connected to the method. How do the two strategies, uni-modal pretraining and uni-modal dropout, improve to reduce complementary information? Only intuitive explanation is proposed. Both direct empirical and theoretical results are not provided.
- In the method section, the authors just jump to the topic of modality laziness, when explaining the method design. What is the correlation between this topic to the missing modality problem. How does this topic influence method design. It needs well explanation.
- In table 1/2, only Miss-V and Miss-A results are shown. It is necessary to well evaluate the method performance under missing modality cases, like testing set with different missing ratios.

 [A] On uni-modal feature learning in supervised multi-modal learning
 [B] Audiovisual SlowFast Networks for Video Recognition

**Questions:**

Most of my concerns are stated in the above sections. Here are some additional questions.
- What is the concrete setting of experiments on Page 8, "UME-MMA on situations where a specific modality of data is missing in the training set". Based on my understanding, the proposed method takes modality dropout as data augmentation strategy, which means that the training data with missing modality is a part of method design. How could it be the advantage that UME-MMA can handle missing modality in training set.

---

### Official Review · Reviewer_v3QK · 2023-11-01

**Soundness:** 3 good
**Presentation:** 2 fair
**Contribution:** 2 fair
**Rating:** 5
**Confidence:** 3

**Summary:**

With the rise of multi-modal learning, research on the resilience of these models in the face of missing modalities is increasing, but previous studies lack comprehensive insight. The paper approaches these situations with information theory and introduces the Uni-Modal Ensemble with Missing Modality Adaptation (UME-MMA), that uses pre-trained weights to enhance feature extraction and adapts better to missing modalities. UME-MMA, built on a late-fusion learning framework, is versatile across various encoders, and their implementation has shown effectiveness in audio-visual and vision-language datasets.

**Strengths:**

- This paper presents a novel approach and interpretation to the multi-modal model with missing modalities, a topic of increasing relevance and applicability.

- It also shows improved performance against several naive baselines, demonstrating the effectiveness of the proposed model.

**Weaknesses:**

- The experimentation conducted in this paper seems insufficient to justify the key ideas. For instance, there is no clear explanation or evidence highlighting the need for uni-modal training due to the multi-modal training dataset's size.

- If a multi-modal data set as large as the pre-trained data is available, it is unclear whether the key components would still operate in the same way.

- The primary performance shown is similar to uni-modal training. The paper suggests that high performance can be achieved simply by late fusion, depending on the presence or absence of modalities in uni-modal conditions. Given that the multi-modal dataset is not extensive, this seems like a fairly common performance result.

- The paper lacks a powerful baseline for situations with missing modalities. It includes only very naive baselines, making it challenging to ascertain the effectiveness of the proposed method.

**Questions:**

- Could you elaborate more on the interlink between the "Uni-Modal Feature Learning in Supervised Multi-Modal Learning" paper and the current study?

- What is the performance outcome when a uni-modal encoder is utilized, and only late fusion layer is trained in a multi-modal setting?

- Why were other related studies on missing modalities not compared in the experiments? How is your approach different or superior to theirs?

---

### Official Review · Reviewer_K5Sq · 2023-11-04

**Soundness:** 2 fair
**Presentation:** 3 good
**Contribution:** 2 fair
**Rating:** 5
**Confidence:** 4

**Summary:**

This paper is about handling missing modalities in multi-modal learning. It analyzes the problem from the infomation theory aspect. Expermentally it shows some improvements over some previous methos on two different kinds of databases, i.e, audio-visual and vusion-langluage.

**Strengths:**

It attempts to show the importance of missing modalities in multi-modal learning case.

It presents the information theory analysis of the missing modality problem.

It shows experiments on two different kinds of databases.

**Weaknesses:**

After the information theory based analysis, it seems that there is not a novel method that is presented to deal with the missing modality problem.

It argues that the key thing is to fully utilize the no-missing modality when handlng the missing modality problem, but it is not clear how to develop a method to achieve this utilization. I tried to find thorough the description in the text, but still have difficulty to get it.

From the experiments, it seems that the presented work is only compared to [Ma 2021] and [Ma 2022], which was from the same group of authors. Not other comparisons to show in the paper. My thought is that (1) the research topic is with narrow interest, since no other groups have shown interests during the period of about 3 years; (2) or may be other methods were not compared with just because of ignorance?

**Questions:**

See my concerns listed in the Weakness part.

---

### Official Review · Reviewer_zEWL · 2023-11-05

**Soundness:** 2 fair
**Presentation:** 3 good
**Contribution:** 1 poor
**Rating:** 1
**Confidence:** 4

**Summary:**

The paper addresses the growing importance of robust multi-modal models in the face of missing modalities. The authors propose an information-theoretic approach to address this challenge. They introduce Uni-Modal Ensemble with Missing Modality Adaptation (UME-MMA), which leverages uni-modal pre-trained weights and data augmentation techniques to enhance feature extraction and adapt to missing modalities. UME-MMA is designed for a wide range of modalities and allows the integration of pre-trained encoders. The paper demonstrates UME-MMA's effectiveness in audio-visual and vision-language datasets.

**Strengths:**

- the paper is well-written and organized, making it easy to follow the authors' arguments.
- the paper provides a theoretical analysis of the interconnection of modality complementariness and robustness in multimodal settings.
- the empirical results clearly demonstrate the proposed method UME-MMA, outperform naive baselines

**Weaknesses:**

- It is hard to see the connection between the theoretical analysis and the proposal method, thus strongly limiting the significance of the theoretical analysis, which is a big component of the paper. Meanwhile, it makes the paper incoherent and with lots of redundancy.
- The paper offers minimal novel insights beyond the intuitive notion that utilizing more complementary modalities can enhance robustness.
- The proposed method is naively simple and the idea behind has been commonly adopted in the literature to improve multi-modal network performance. However, such works are not acknowledged and cited in the paper.
- In terms of experiments, in many cases, the proposed method does not perform better than uni-modal models.
- Also, regarding robustness, only a simple baseline is compared, and almost no ideas/methods from previous works have been compared to, for example: https://arxiv.org/pdf/2204.05454.pdf and https://arxiv.org/abs/2303.03369

**Questions:**

N/A